



# Reliable reference for the methane concentrations in Lake Kivu at the beginning of industrial exploitation

Bertram Boehrer[1], Wolf von Tümpling[1], Ange Mugisha[2], Christophe Rogemont[3], Augusta Umutoni[4]

[1] Helmholtz-Centre for Environmental Research – UFZ, Magdeburg, Germany
[2] Lake Kivu Management Program LKMP, Gisenyi, Rwanda
[3] Hi-Tech Detection Systems, Massy Cedex, France
[4] Lake Kivu Management Program LKMP, Kigali, Rwanda

*Correspondence to*: Bertram Boehrer (Bertram.Boehrer@ufz.de)

**Abstract.** Dissolved methane in Lake Kivu (East Africa) represents a precious energy deposit for the neighbouring countries, but the high gas loads have also been conceived as a threat by the local population. This is especially the case when stratification in the lake is changed during the planned industrial exploitation. Both issues require accurate and reliable measurements of dissolved gases and temporal changes to take responsible action. Previous data fulfilled these requirements only unsatisfactorily. Prior to our measurements, there was considerable disagreement about prognosticated new formation of methane. We show how measurement accuracy could be significantly improved by implementing equipment, which was especially designed and modified for the complex gas conditions in Lake Kivu. From 150m to 430m depth, samples were taken to determine the amount of dissolved methane and dissolved carbon dioxide more reliably and more accurately. Beyond the provision of gas concentration profiles at the beginning of exploitation, this investigation should also provide methods to survey the further evolution of gases over time. The use of gas tight sampling bags produced highly reliable and accurate measurements. Our measurements confirmed the huge amount of stored methane, but they do clearly not support the current believe of a significant recharge beyond diffusive loss. Direct measurements with a custom-made gas pressure sensor indicated no imminent endangerment through limnic eruptions. A further survey of gas pressures, however, is mandatory to detect changing conditions. With sampling bags and gas pressure sensor, we introduced reliable and highly accurate measuring approaches for the survey of the further development of gas concentrations. This equipment only requires little effort for calibration, which can also be accomplished in remote areas of Africa.



# 1 Introduction

Lake Kivu, located on the border between Rwanda and the Democratic Republic of the Congo in Central and East Africa, contains large amounts of dissolved methane ($CH_4$) in its permanently stratified deep water (Tietze 1978, Tietze et al. 1980, Schmid et al. 2005). It hence represents an important resource for the neighbouring countries, especially Rwanda, which currently has no access to any other hydrocarbon deposit in its territory. The commercial-scale exploitation of this resource was started on 31 December 2015 with the commencement of the KivuWatt power plant operation Phase 1 with an installed capacity of 26 MW. Further power plants are planned on both sides of the border. For a responsible handling, observation and the survey of the management prescription, a reliable reference of gas content and the development of suitable measurement equipment for documenting its temporal evolution are mandatory.

In addition to methane, large amounts of carbon dioxide are dissolved in the deep water. Together, these gases potentially pose a risk for the local population, as spontaneous degassing could be feared, where large amounts of gas could be released from the lake with catastrophic consequences for the local population as it happened at Lake Nyos in 1986 (Sigurdson et al. 1987, Kling et al. 1987). Previous measurements indicated rising gas concentrations in Lake Kivu and hence an increasing risk over time scales of few decades (Schmid et al. 2005). To avoid any endangerment for the population, a management prescription for withdrawal depth and deposition of partially degassed deep water and wash water needed to be developed to avoid damage to the lake ecology and endangerment of the local population.

There have been a number of measurement campaigns dating back as far as 1935 aiming at quantifying the methane deposit in Lake Kivu (Damas 1937, Schmitz and Kufferath 1955, Degens et al. 1971, 1973, Tuttle et al. 1990). Either samples were locked in containers and recovered to the surface or hoses were used to bring deep water in a continuous flow to the surface. All of them had to struggle with the loss of gas and water while recovering samples even the more recent measurement trials (Tassi et al. 2009, Pasche et al. 2011). So far, only one published measurement series used in-situ sensors (Schmid et al. 2005).

Currently, the gas content of Lake Kivu is mainly based on the gas concentrations measured in November 2003 by the team of Michel Halbwachs (published in Schmid et al. 2005). The temporal evolution, i.e. an assumed recharge with gases has been quantified from the difference to measurements by Klaus Tietze in 1974/5: Schmid



et al. (2005) concluded that $CH_4$ concentrations increased by up to 15% within three decades, potentially leading to an increased risk of a gas eruption. According to this prognostication, action to release the gas pressure would have been required to avoid an endangerment of the local population. However based on a more detailed analysis of the carbon budget of the lake, Pasche et al. (2011) concluded that the concentrations most likely were not

increasing as fast.

In conclusion, the data availability was not sufficient to take responsible measures for the exploitation of the gas resource and the risk assessment during exploitation. Hence, the Rwandan government took action to invite specialist teams to implement their measurement approaches in Lake Kivu to get a data base reliable enough for

political decisions (e.g. Wüest et al. 2012). Various approaches hence were modified for the special conditions in Lake Kivu, each requiring considerable effort to meet the expectations (Schmid et al. 2019). In this paper, we present gas measurements from waters that have been collected in sampling bags and an analysis of the gas composition by gas chromatography. An intercomparison with competing approaches (Grilli et al 2014, 2016) will be published in separate where also the complex conversion between gas pressures and gas concentrations

will be done.

We felt that, for the special case of Lake Kivu, a reliable measuring method must be implemented that is suited for the local scientific staff to document changes in the gas charge at a later time. Only measuring techniques should be included that required few calibration that could be reliably done at a remote location as Lake Kivu.

The approach should be comprehensible and avoid any hidden errors. In conclusion, we modified the sampling method from the gas charged mining lake Vollert-Sued (Horn et al. 2017) and Guadiana pit lake (Sanchez-Espana et al. 2014, Boehrer et al 2016) for the conditions in Lake Kivu. In addition to methane and carbon dioxide concentration, we also included measurements of dissolved solids in this paper. Finally, we measured total dissolved gas pressure (TDGP) using a customized sensor, since TDGP is the appropriate measurement for

judging the proximity to spontaneous ebullition.



## 2 Study site, methods

### 2.1 Lake Kivu

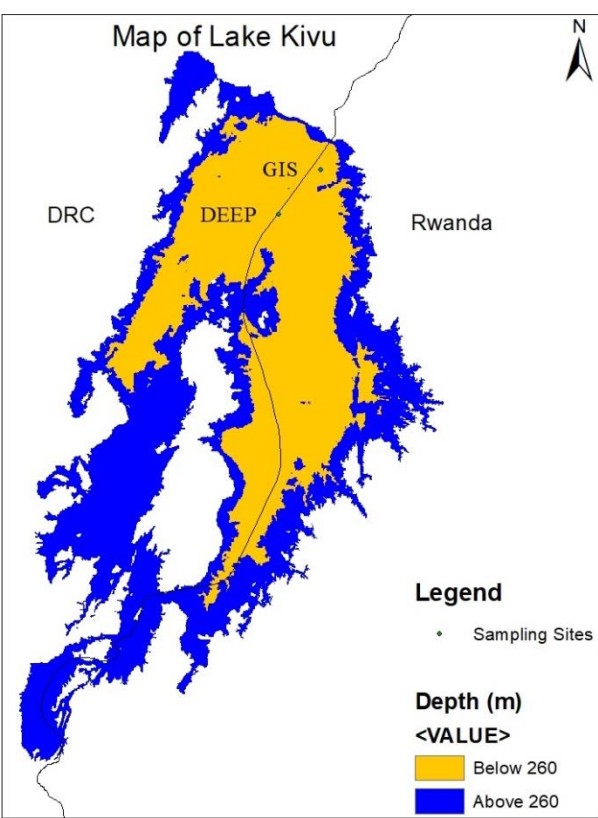

**Figure 1: Lake Kivu map marking the area of the resource zone below 260 m depth within Lake Kivu indicating the location of sampling site GIS and DEEP**

The measurement campaign took place from 9 to 13 March 2018 near Gisenyi/Rubavu at the Northern shore of Lake Kivu in Rwanda. Sampling was mainly accomplished from the convenient sampling platform ("GIS" in Fig.1) at 1.74087°S / 29.22602°E about 5 km distance from Gisenyi at a maximum depth of 410 m, while the deepest sample was taken close to the border to Democratic Republic of the Congo at the deepest location ("DEEP" on Fig. 1) on Rwandan territory 1.79865°S / 29.17172°E to extend the sampling range as deep as possible. The horizontal distance between the sites and the different sampling dates were not expected to influence comparability, as horizontal transport was much faster than vertical transport and gas production or



consumption, as indicated by (Schmid and Wüest 2012, Ross et al. 2015). Therefore, it was reasonable to assume horizontal and temporal steady-state for the duration of the campaign. In addition, a few samples from an earlier campaign in 2017 were included in this paper.

## 2.2 Multiparameter profiles

Vertical profiles of temperature, electrical conductivity, oxygen concentration, pH and turbidity were measured at the platform and the deep location using a multiparameter probe CTM1143 (Sea and Sun Technology, Germany). Sensor properties and description can be found on www.sea-sun-tech.com.

Here and in all following methods, pressure (bar) was converted to depth (m) by dividing through 0.0978 bar m$^{-1}$. Temperature compensation of electrical conductivity to 25°C was done using the equation

$$C_{25} = \frac{C(T)}{0.0194 \cdot (T - 25°C) + 1} \qquad (1)$$

In 2018, the oxygen zero point was slightly off calibration (by +0.37 mg/l), which was corrected after sampling.





## 2.3 Water sampling for gas analysis

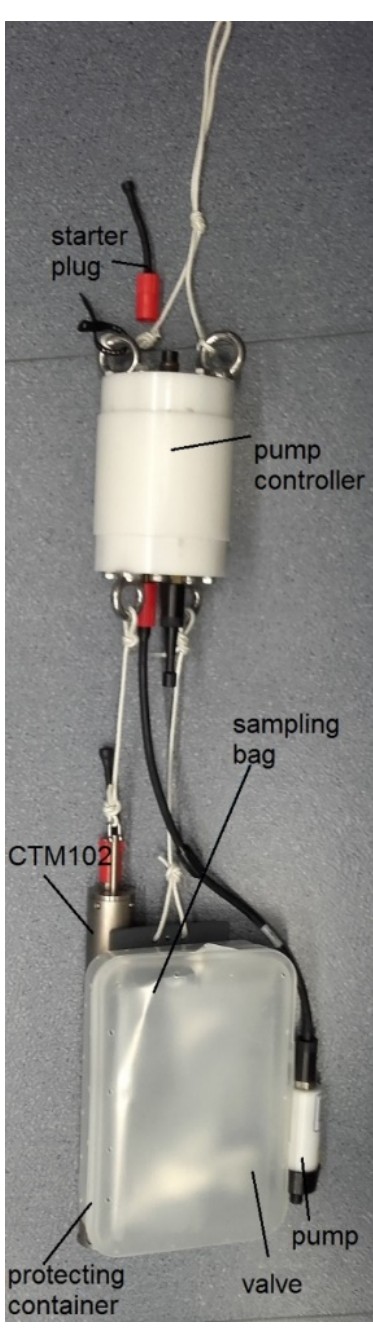

**Figure 2: Gas sampling arrangement as used in Lake Kivu with pump controller, submersible pump, and protection housing for the sampling bag; a small CTD-probe was connected to record depth, electrical conductivity and temperature and started**

5 **synchronously with the pump controller using a starter plug.**



For gas measurements, gas tight sampling bags (TECOBAG; see Horn et al 2017) were lowered to the investigation depths and partially filled with water by operating a pump for a short period. Enough remaining capacity of the bag was retained to accommodate the total amount of gas when bags were recovered and thus pressure was reduced to atmospheric level. Other than in previous implementations of this technique in Lake

5    Vollert-Sued, Germany (Horn et al. 2017) and Guadiana Pit Lake, Spain (Boehrer et al. 2018), the pumps were switched on and off by a submersible controller (see Fig. 2). For all samplings, a CTD probe (Sea and Sun Technology; CTM102) accompanied the sampling equipment for an accurate sampling depth determination. After filling, sampling bags were transported to the LKMP laboratory and left there over night to equilibrate gases between gas phase and liquid phase.

The volume of water was measured by weighing the bags on the laboratory weight scale of LKMP (subtracting the weight of bags) and dividing by density (e.g. Moreira et al. 2016). The volume of the gas space was measured thereafter by withdrawal through syringes from the bags. Part of the gas withdrawn from the sampling bags was introduced into a gas chromatograph (GC, Perkin-Elmer Clarus 580) in the LKMP laboratory to detect the gas

composition quantitatively. Calibration was performed with dry gas standards of composition 20% $N_2$, 40% $CH_4$, 40% $CO_2$ at an accuracy of 0.7% or better. In all samples, the sum of all detected gases amounted to around 97%. The remainder of undetected 3% corresponded very well with the expected moisture at laboratory temperatures and hence were attributed to water vapour. The measured concentrations of $CH_4$ and $CO_2$ were multiplied by the gas volume to yield the amount in the gas space. To determine the entire amount of $CH_4$ in the sample, the

residual portions dissolved in the water were calculated assuming equilibrium between the water and gas phase at temperature and pressure in the laboratory. As the gas volume was of the same order of magnitude as the water inside the bag, this contributed a few percent to the total amount of $CH_4$.

Due to its high solubility however, a considerable portion of $CO_2$ remained in solution, and hence required a

more accurate determination. Only in the case of $CO_2$, the dependence of the Henry coefficient on electrical conductivity (of few percent in the considered range) was included in the calculation: based on electrical conductivity $C_{25}$ at the sampling depth (see Fig. 2) the Henry coefficient was interpolated between freshwater ($C_{25} = 0$ mS/cm) and seawater ($C_{25} = 53$ mS/cm) (given by Sander 2015, Murray and Riley 1971, see also Boehrer et al. 2016). In addition, the concentration of bicarbonate in the water phase was calculated based on pH-

measurements in the laboratory by inserting a sensor after the volume and GC measurements were completed. A dissociation constant of $pK_1 = 6.2$ was used (Cai and Wang 1998). Hence, we could evaluate the dissolved

inorganic carbon DIC as the sum of three contributions: $CO_2$ in gas space, $CO_2$ dissolved in water, and $HCO_3^-$ ; $CO_3^{2-}$ is practically not present at the given pH. $CO_2$ concentrations in the lake were calculated by adding the $CO_2$ gas volume in the sampling bag and the dissolved amount. The tiny contribution coming from bicarbonate shifting to dissolved $CO_2$ could be quantified (relative contribution of $10^{-5}$) on the base of pH change from the

field (multiparameter profiles) to the laboratory measurement. Hence it could have been neglected without increasing the expected error.

The accuracy of the measurements was estimated from the following contributions: accuracy of volume measurement of gas 4% (above 250 m 6 % due to smaller volume); Henry coefficients for fresh water (and salt

water) were known within 5%, temperature fluctuations in laboratory allow for less than 5% plus unknown variation of the Henry coefficient with Kivu-salts less than  5%; added as independent errors. This resulted in an 8% error for the Henry coefficient. As less than half of the $CO_2$ remained in solution, this contributed less than 4% to the expected error. In the case of $CH_4$, only a small fraction remained in solution, and the corresponding error contribution was less than 1%. The error in the measurement of the mass (and volume) of the sample was

smaller than 1%. Altogether, we expected a precision of the measurement of 5% for $CH_4$ (7% above 250 m depth) and 6% for $CO_2$ (8% above 250 m). If required, the accuracy could be improved by implementing a better volume measurement of the gas. In the case of $CO_2$, a more exact knowledge of the Henry coefficient would help as well.

In conclusion, the sampling bags represented a simple approach, which required careful sampling and a good understanding of the solubility of gases for the processing of data. Results, however, were reliable as the approach did not provide much space for hidden errors. Gas tight sampling bags for $CH_4$ and $CO_2$ were a precondition, which was very well accomplished for the investigated gases in the bags used (see Horn et al. 2017).

## 2.4 Total gas pressure

For gas pressure measurements, we used a Pro-Oceanus probe, which was especially customized for the

application in Lake Kivu, as standard sensors for total dissolved gas pressure (TDGP) were not suited for the


deployment depths and/or expected gas pressures in Lake Kivu. The pressure sensor contained a small gas measurement volume separated from the lake water by a membrane, which was permeable for dissolved gases such as $CH_4$, $CO_2$ and $N_2$. The partial pressures of all gases in the measurement volume adjust to equilibrium with their concentrations in the water. If the total gas pressure inside the gas space equals the outside pressure

(hydrostatic plus atmospheric pressure), a virtual gas bubble withstands the pressure at depth and persists long enough to start moving upwards through the water column through its own buoyancy. Hence, the ratio of the total pressure inside the measurement volume compared to absolute pressure represents a quantification of the proximity of lake water to spontaneous ebullition (e.g. Schmid et al. 2004, Koschorreck et al. 2017).

Before deployment, the sensor needed to be immersed in water for several hours at a total pressure above surrounding gas pressure. Thereafter, a response time of only few minutes was expected. Measurements were performed from the platform at discrete depths where the probe was left for about 20 to 30 min (Fig. 3). Data were recorded continuously, and from the time series at 286.1 m depth, a response time of less than 4 min (half-value time $t_{1/2}$ of 150 s) was determined by fitting an exponential curve to the observations. As the available time

for measurements was limited and also the recovery of the sensor required time, measurements were done at seven discrete depths. The CTM1143 probe accompanied the TDGP sensor for an accurate depth reference.

In order to compensate for the measured response time, the total gas pressure TDGP at time $t$ was calculated from the measured pressures $p_{meas}$ at the times $t$ and $t - t_{1/2}$ .

$$TDGP\,(t) = 2\,p_{meas}(t) - p_{meas}\big(t - t_{1/2}\big) \qquad (2)$$

TDGP approached the final value much faster than the original readings $p_{meas}$ . The expected error of the measurement was estimated from calibration error ( <0.5% of actual measurement) plus 0.04 bar uncertainty from reading.

## 2.5 Trace element measurements

Samples were collected in a standard grey PVC Niskin Sampler with a compensation reservoir made from

30 synthetic rubber from Ocean Scientific International (OSIL), United Kingdom, was used and partially filled in 50 mL Sarstedt tube. A second part was stored in 250 mL PE bottle for ca. 4 h. After arriving the LKMP laboratory





a second 50 mL Sarstedt tube was filled with the syringe filtrate through 0.45 µm membrane filters (Millipore). For preservation, samples were acidified (pH < 2) by adding 250 µL nitric acid 65% (Merck suprapur) to 50 mL samples. 20 mL of unfiltered water were pressure- and temperature-controlled microwave digested (CEM discover) in quartz tubes with 1 mL $H_2O_2$ 35% (Supelco) and 1.5 mL $HNO_3$ 65% (Merck suprapur). For the

5    acidified filtrate, no further sample preparation was necessary. Trace elements were determined by inductively coupled plasma MS (ICP-MS/MS; Agilent 8800, Agilent Technologies, Germany) according to the norm (DIN EN ISO 17294:2017-01). As a measure for quality assurance, the NIST reference material SRM 1643e - Trace Elements in Water was used. Recovery rates between 95 and 107 % were achieved.

10   K, Na, Mg and Ca concentrations were measured from aliquots of the digested sample with Perkin Elmer Optima 7300 DV inductively coupled plasma optical emission spectrometer (ICP-OES) under the application of the norm (EN ISO 11885:2009). Dissolved $SO_4^{2-}$ and $Cl^-$ anions have been determined by liquid chromatography (Dionex ICS 3000) as defined in the norm (EN ISO 10304-1:2009) after isocratic separation. The TOC was quantified with a DIMATOC® 2000 from Dimatec Analysentechnik Ltd., Essen, Germany using the combustion-infrared

method described in the Guidelines (EN 1484-1997). Bound nitrogen (TNb) was measured by wet chemical per sulphate digestion according to Koroleff followed by nitrate reduction to detect the nitrogen as nitrite photometrical with a segmented flow analyser (SFA) from Skalar, Dutch instrument manufacturer (EN ISO 11905-1:1998). Total phosphorus (TP) was detected as phosphate after wet chemical oxidation with the Ammonium molybdate spectrometric method on a Hach photometer DR 5000 (EN ISO 6878:2004). For quality

assurance, the NIST reference material SRM 1643e - Trace Elements in Water was used. Recovery rates between 95 and 107 % were achieved. Annual round robin tests for all analysed parameters from r-concept had confirmed the applicability of the methods.





# 3 Results

## 3.1 Multiparameter profiles

The profile on 13[th] March 2018 showed typical conditions for the wet season, where the surface layer of the lake

5   (i.e. the top 60 m, which can undergo seasonal mixing during the dry season) is thermally stratified (Fig. 3). There was a steep oxycline with strongly decreasing oxygen concentrations between 25 and 40 m depth, and below about 45 m depth, the water column was anoxic. Below 60 m depth in the monimolimnion of this meromictic lake (Boehrer and Schultze 2008, Gulati et al 2017, Boehrer et al. 2017), the profile showed the usual stepwise increase in temperature and conductivity and decrease in pH as documented in earlier measurements.

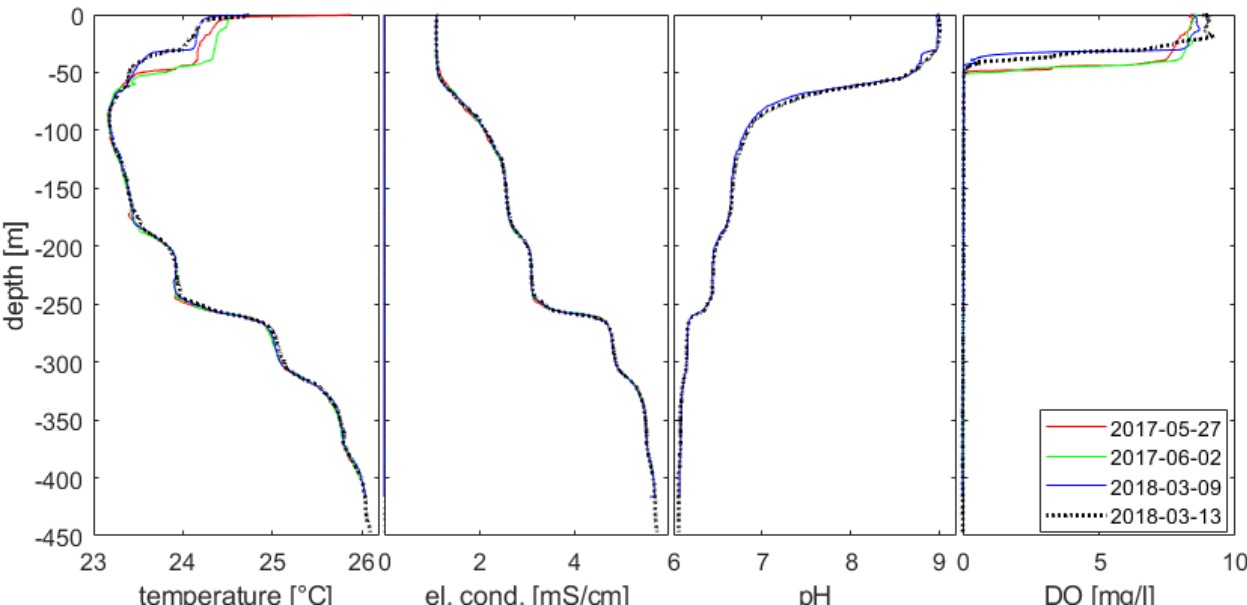

**Figure 3: Profiles of in-situ temperature, electrical conductivity (temperature compensated for 25°C), pH (only 2018), dissolved oxygen concentration and turbidity of Lake Kivu water against depth. All profiles were measured with the same multiparameter**
15   **probe (CTM1143, Sea and Sun Technology, Germany) from the platform "GIS" except for the profile on 13[th] March 2018, which was acquired close to the deepest location "DEEP" on Rwandan territory.**



In the deep water, the biggest gradient in temperature and electrical conductivity at 260m represents the upper limit of the resource zone. Below, concentrations are high enough for industrial exploitation. pH values, which are crucial for DIC calculations, are nearly identical to values shown in the work by Schmid et al. (2005).

## 3.2 Methane and carbon dioxide

Thirteen depths were sampled successfully in March 2018. In addition, five samples from a previous field campaign on 27[th] May and 2[nd] June 2017 were included in the analysis. The samples covered the depth range
10 from 150 m to 432 m depth. The measurements confirmed the extremely high gas concentrations below 260 m depth ("resource zone") for both methane and carbon dioxide (Fig. 4).





**Figure 4: Concentrations of methane (upper panel) and carbon dioxide (middle panel) with uncertainties against depth in Lake Kivu. Samples were acquired in May/June 2017 (squares) and March 2018 (circles) at the platform, except for the deepest sample which originated from Site 2; TDGP measurements were measured at site 1 (lower panel: error bars are smaller than symbols).**
5 **The solid line represents absolute pressure (hydrostatic plus atmospheric).**

### 3.3 Total gas pressure

Total dissolved gas pressures increase with depth, reaching more than 17 bars at depths greater than 400m (Fig. 4). Parallel to the measured gas concentrations, the major gradient lay between 230m and 290 m, i.e. between the potential resource zone and the resource zone and hence followed the shape of the methane and the carbon dioxide curve. At any depth, TDGP was much lower than absolute pressure. Within the resource zone, a vertical excursion of at least 150m would be required for a water parcel to start spontaneous ebullition. At about 350 m depth, gas pressure reached about 17 bar, which was about half the absolute pressure. Gas pressures would need to double to form bubbles spontaneously.

### 3.4 Trace element measurements

Measurements of trace elements are listed as total and dissolved (filtered through 0.45µm) concentrations (Table 1). Main ions and nutrient concentrations are listed in Table 2.

*(Table 1 see below)*

**Table 1: trace element concentration in Lake Kivu: total and dissolved (0.45 µm filtered, lower part of the table); all concentrations are in [µg/l]; depth in [m of rope], which to our experience was 6% higher that real depth. Water was sampled from platform (Site 1) on 8.03.2018; only the depths 410 m and 450 m were sampled from the boat (deep) on 12.03.2018.**

*(Table 1 continued see below)*

**Table 1 (continued for more elements)**

*(Table 2 see below)*

**Table 2: main ions and nutrient concentration in Lake Kivu; all concentrations are in [mg/l]; depth in [m of rope], which to our experience was 6% higher that real depth. Water was sampled from platform (Site 1) on 8.03.2018; only the depths 410 m and 450 m were sampled from the boat (deep) on 12.03.2018.**





## 4 Discussion

### 4.1 Stratification

The profile measurements of temperature, el. conductivity, oxygen and pH confirmed the picture that was known

5 from previous measurements. Seasonal variability extended down to about 60m depth, which could be seen in some variability of temperatures. Up to this depth, the recirculation also supplied dissolved oxygen, while below, a very constant picture was sustained: the stepwise structure of the deep water (monimolimnion), which could be clearly recognized by sharp changes in temperature and el. conductivity. This concurred with our expectations from the estimated renewal time of close to 1000 years in the deeper monimolimnion (Schmid and Wüest 2012).

The comparison between profiles from 2017 and 2018 confirmed this picture: the seasonal dynamics was restricted to the upper 50 m, where difference in temperature and temperature stratification could be detected. The oxycline showed vertical excursions from about 45 m depth (May 2017) to about 30m depth in March 2018. Below 50m depth, the measurements showed no temporal variability in temperature, electrical conductivity in

15 neither pH nor oxygen. Small changes in the temperature profile appeared at around 180m. Beyond this, the measurements confirmed the situation as it was known from earlier campaigns.

### 4.2 Comparison with earlier gas measurements

Measured concentrations of $CH_4$ and $CO_2$ were compared with previous observations by Schmitz and Kufferath from 1952/4, K. Tietze from 1974/5, by M. Halbwachs and J.-C. Tochon from 2003 (published in Schmid et al 2005), and by Schmid et al. 2004 (Published in Schmid et al. 2005). For the management of the lake, both considering commercial $CH_4$ extraction and the risk of a gas outburst, the most important depth range was the

25 resource zone below the main gradient. In this range, the concentrations from this campaign were between those measured by K. Tietze in 1974/5 and those measured by M. Halbwachs and J.-C. Tochon in 2003 (see Fig. 5).



For $CO_2$, the concentrations agreed well with previous measurements. Above 260 m and below 400m, the measurements lay very close to the Tietze and the Halbwachs data; only between 260m and 380m our data showed lower values (Fig. 6) with differences marginally above the expected accuracy of this campaign.

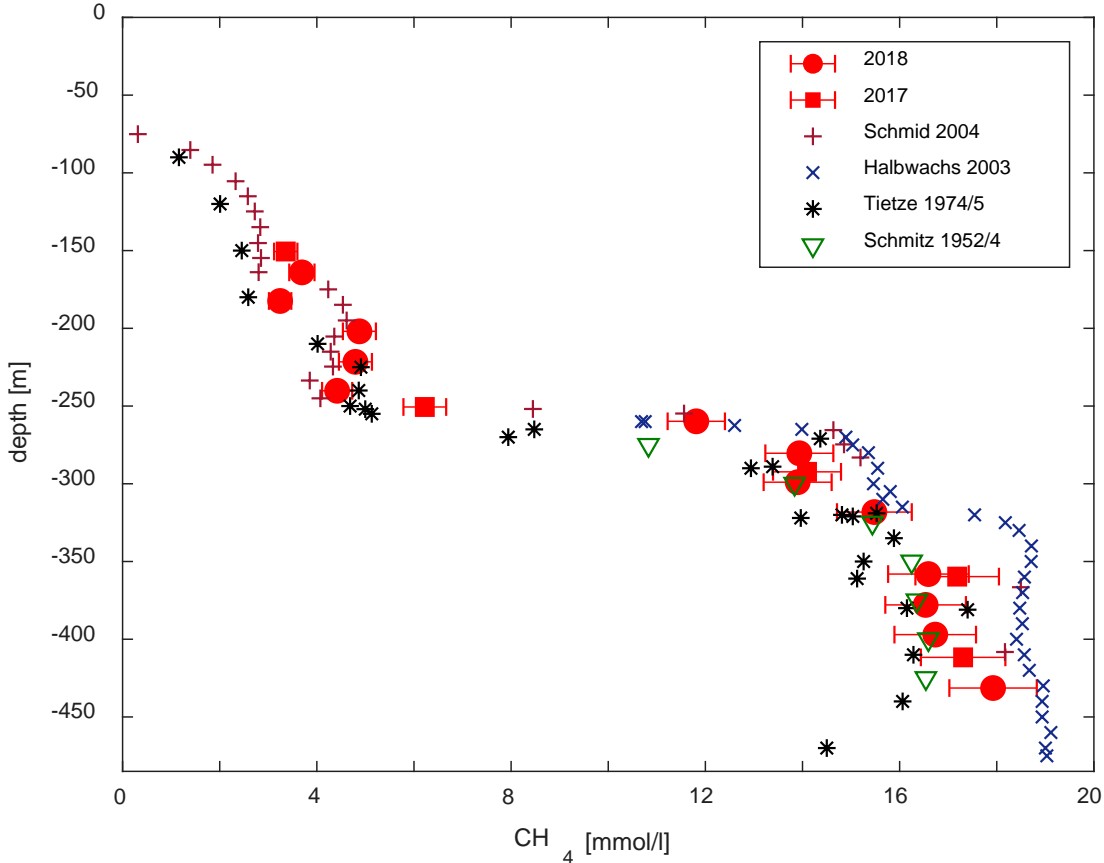

**Figure 5: Methane concentration from this study compared to previously published observations (numerical values of previous measurements are taken from [14]).**

The new measurements did not confirm an increasing trend of methane concentrations in Lake Kivu (as reported
10  in Schmid et al. 2005, see Fig. 5). Though the observed concentrations lay marginally above measurements by K. Tietze in 1974/5, they were significantly lower than the measurements by M. Halbwachs and J.-C. Tochon in 2003. To our conviction, an increase of methane concentrations over decades followed by a decrease would not represent the true dynamics of $CH_4$ in the deep water. Such a variability would be questioned by the residence





time of water in the resource zone of nearly 1000 years (Schmid et al. 2012) and the constant electrical conductivity profiles. In addition, the lower concentrations in the present study could not be caused by the KivuWatt gas extraction as the volume of removed water was too small and the withdrawal would result in a thinner resource zone but not in lower concentrations. Our observations could neither confirm the methane production in the resource zone of 93 g C m$^{-2}$ yr$^{-1}$ (0.18 km$^3$ yr$^{-1}$) as postulated by Pasche et al. (2011), which would correspond to a growth in concentrations by approximately 0.3% per year. In conclusion, temporal changes in gas concentrations lay below the detection limit at the accuracy of the current and the previous gas measurement methods.

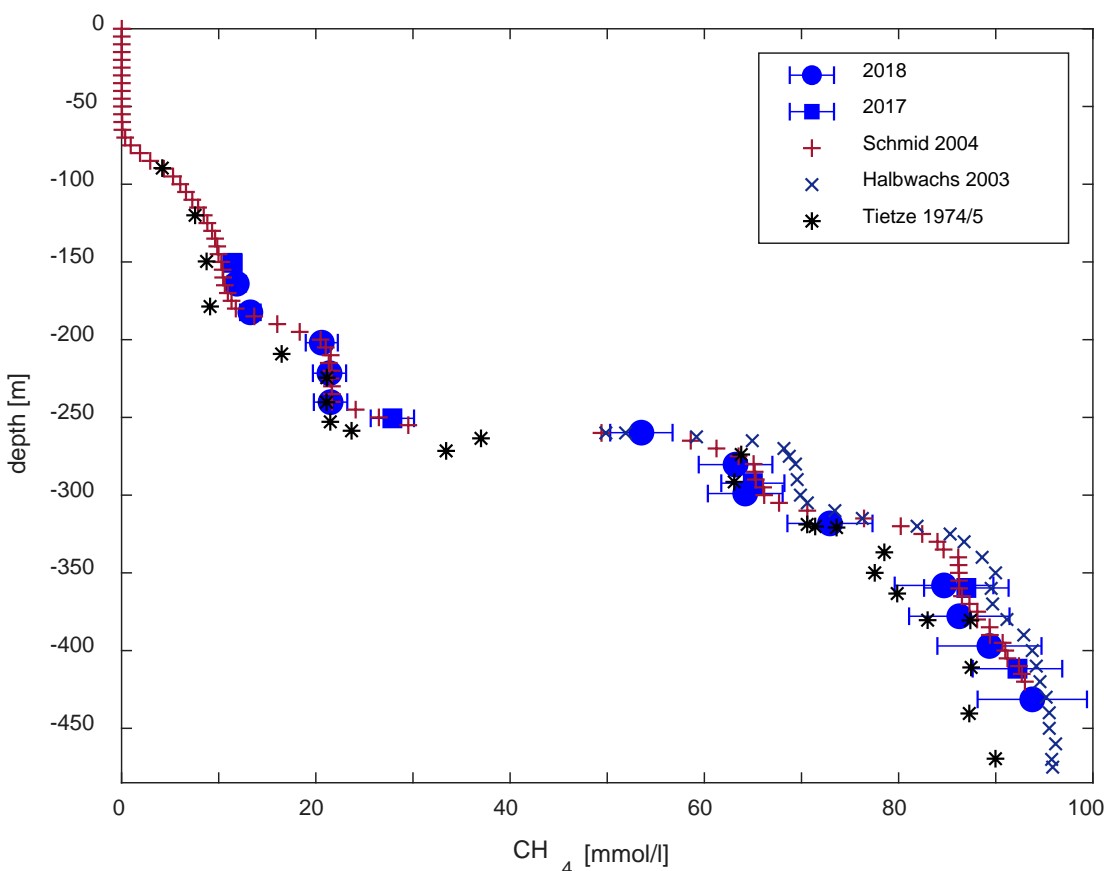

**Figure 6: Carbon dioxide concentration from this study compared to previously published observations (numerical values of previous measurements are taken from [14]).**

However, a certain amount of methane had to be produced to balance the continuous loss from the lake by upward transport and subsequent oxidation in the surface layer or (to a small extent) emission to the atmosphere (Pasche et al. 2011, Borges et al. 2011), if concentrations remained constant. This amounted to 32 g C m$^{-2}$ yr$^{-1}$ (0.06 km$^3$ yr$^{-1}$ in the resource zone and 0.013 km$^3$ yr$^{-1}$ in the potential resource zone) for steady-state. Pasche et al. (2011) estimated it marginally higher at about 35 g C m$^{-2}$ yr$^{-1}$.

## 4.3 Total gas pressure

The in-situ measurements of TDGP indicated that spontaneous ebullition from the water will not happen in near future: the difference from the measured gas pressure to the absolute pressure at any depth showed a big safety margin. As also the gas concentrations had not changed significantly over the last decades, an abrupt increase towards dangerous conditions was not indicated by our data.

The Oceanus-Pro custom-made sensor for Lake Kivu showed a convincing performance. After proper pre-conditioning, measured response time based on in-situ data from Lake Kivu lay just below 4 min (half-value time was 150 s). This was faster than any previously published similar measurement, but still too slow for straight profiling. However, we could prove that sufficient data points could be measured in the resource zone (and the potential resource zone) with acceptable effort. Such a spatially better resolved profile would be feasible and would be highly desirable as reference for the temporal evolution.

In addition, the accuracy of the TDGP measurement only depended on the calibration of the pressure sensor, which could be accomplished at high accuracy. Hence this direct in-situ measurement would probably be the first to indicate changes in gas concentrations, if proper measurements were done at regular intervals. In addition, such gas pressure measurements are the proper indicator for endangerment by limnic eruptions and hence could serve both purposes.

Gas pressure had to be attributed to volatile solutes. The major contributors were methane, carbon dioxide and nitrogen. While we had good measurements of methane and carbon dioxide, we missed data of similar quality for



nitrogen: hence, we could not give a proper calculation of gas pressure from gas concentrations. In addition, gas pressures at extreme concentrations as in Lake Kivu and high absolute pressures would show a complex behaviour (see fugacities), and an accurate conversion of gas concentrations to gas pressures would go beyond the purpose of this paper. Hence we decided to publish this connection in a separate paper, but we felt the need

for some rough estimates as a plausibility check for the measurements.

A quick calculation, based on Henry coefficients at normal pressure, attributed about 12 bar gas pressure to methane at 400m depth and about 2.6 bar to carbon dioxide; nitrogen was unknown, but expected somewhere between 0 and 1 bar (~0.68 bar at equilibrium with atmosphere at Kivu altitude). Other contributions like water

~0.03 bar and Argon ~0.01 bar were small. This added up to about 15.4 bar. The proper calculations of fugacities will correct these values in the range of 10 % possibly 20%. The direct measurement from the Oceanus-Pro sensor showed 17.227 bar. This confirmed the correct order of magnitude for the measurements.

## 4.4 Trace metals

In general, our chemical data did not show any values of concern. Comparing our data with previous (Tassi et al. 2009), we found no significant differences for Al, Ba, Co, Mn, Nb, Ni, Rb, Sr, and Y. The same was true for the determined main ions Na, K, Mg, Ca, Cl, $SO_4$ and $NH_4$. Observed very high Chromium concentration with up to

447 µg/L in Kabuno bay and 281µg/L in the main basin (Tassi et al. 2009) were not confirmed by our data (ca. 1000 times lower) and the difference has to be attributed to contamination or inaccuracies during data processing of the previous measurements (Tassi et al. 2009). In conclusion, the Chromium limit for drinking water (of 50 µg/L in Germany according to the standard, TrinkW) was not exceeded. For Copper, we generally found 10 to 100 times lower concentration than Tassi et al. (2009). Our measured Iron values were often 10 to 20 times

lower. In addition, the numerical values of further investigated elements were measured for dissolved and total element concentration .

# 5 Conclusions

We could present a sampling method for dissolved gases in Lake Kivu, which outperformed all other previously applied techniques in relation to accuracy, applicability and reliability. The required equipment could be cheaply
purchased and implemented by the local scientific personnel in the existing facilities. The current accuracy (of 5 % for methane and 6% for carbon dioxide in the deep water) was sufficient for most purposes, but could be easily and significantly improved by a better volume measurement.

The new measurements of dissolved gases fell within the range of previously published data. The methane
measurements showed slightly higher concentrations than what was measured in 1974 (Tietze 1978, Tietze et al. 1980) but significantly lower than measurements from 2003 by Halbwachs and 2004 by Schmid et al. (2005). The stratification of deep water showed no significant changes since scientific investigations of corresponding accuracy have been documented. Temperature, electrical conductivity, oxygen and pH in the deep water showed only little temporal variability. Only the upper 50 m showed dynamic seasonal behaviour. Measurements of trace
elements did not show any concentrations of concern.

Rising methane concentrations as postulated on the base of earlier measurements were clearly not happening at rates estimated previously. Hence once exploited, a further recharge at a considerable rate is questionable on the base of the current data. If concentrations are rising, the rates are so low that they cannot be quantified due to the
limited accuracy of previous (and the current) gas measurements. A comparison with earlier data suggests that Lake Kivu has been close to a dynamic equilibrium, where newly produced methane replaces the amount continuously lost by diffusion (and mixing) to shallower layers of the lake. The current measurements do not indicate the necessity of expanding exploitation to prevent limnic eruptions.

For safety assessment, direct measurements of gas pressure are feasible with the used prototype or instruments of similar design. These data state quantitatively how large the safety margin to possible spontaneous ebullition is. The presented sensor is fast enough that high resolution profiles are feasible in an acceptable time frame. As pressure sensors can be calibrated at high accuracy, direct gas pressure measurements offer the fastest perspective to detect changes in gas load of Lake Kivu.



**Acknowledgements**

The authors would like to thank 1) Martin Schmid for information before the field trips, for help during data evaluation and the proliferation of data of previous measurements both methane and carbon dioxide measurements from Schmid et al. (2019). 2) Fabian Bärenbold for critical inspection of methane and carbon dioxide data processing 3) Maximilian Schmidt for the collaboration in field and air pressure measurement 4) Roberto Grilli and 5) Francois Darchambeau for the collaboration during the field campaign 2018. 6) Technical staff of UFZ and LKMP. Mark Barry of PRO-OCEANUS for providing a custom-made gas pressure sensor for Lake Kivu.

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





**Tables:**

**Table 1: trace element concentration in Lake Kivu: total and dissolved (0.45 µm filtered, lower part of the table); all**
5 **concentrations are in [µg/l]; depth in [m of rope], which to our experience was 6% higher that real depth. Water was sampled from platform (GIS) on 8.03.2018; only the depths 410 m and 450 m were sampled from the boat (DEEP) on 12.03.2018.**

| depth | Ag | Al | As | Au | B | Ba | Be | Bi | Cd | Co | Cr | Cu | Fe | Mn |
|---|---|---|---|---|---|---|---|---|---|---|---|---|---|---|
| 0 | <0.01 | <20 | 1.6 | <0.01 | 78 | 35 | < 0.1 | <0,01 | <0.01 | 0.04 | < 0.5 | 0.7 | <10 | <7 |
| 10 | 0.035 | 38 | 2 | <0.01 | 172 | 141 | < 0.1 | <0,01 | <0.01 | 0.1 | 0.7 | 3.6 | 60 | 345 |
| 70 | <0.01 | <20 | 2 | <0.01 | 193 | 136 | < 0.1 | <0,01 | <0.01 | 0.09 | < 0.5 | 1.4 | 79 | 417 |
| 130 | <0.01 | <20 | 2.3 | <0.01 | 436 | 443 | < 0.1 | <0,01 | <0.01 | 0.18 | < 0.5 | 0.3 | 18 | 366 |
| 170 | <0.01 | 103 | 2 | <0.01 | 1850 | 438 | 0.1 | <0,01 | <0.01 | 0.16 | < 0.5 | 1.1 | 22 | 328 |
| 230 | <0.01 | 24 | 2.3 | <0.01 | 576 | 595 | 0.17 | <0,01 | <0.01 | 0.2 | < 0.5 | 0.8 | 10 | 307 |
| 290 | 0.015 | 37 | 3.2 | <0.01 | 921 | 784 | 0.4 | <0,01 | <0.01 | 0.35 | 0.5 | 0.5 | n. a. | 301 |
| 350 | 0.021 | 40 | 3.8 | <0.01 | 963 | 747 | 0.43 | <0,01 | <0.01 | 0.54 | 0.6 | 0.3 | n. a. | 281 |
| 410 | 0.021 | 31 | 3.3 | <0.01 | 763 | 712 | 0.31 | <0,01 | <0.01 | 0.5 | 0.6 | 1.2 | n. a. | 295 |
| 450 | 0.012 | 29 | 2.6 | <0.01 | 489 | 566 | 0.28 | <0,01 | <0.01 | 0.32 | 1 | 0.7 | 22 | 286 |
| Dis-solved | Ag | Al | As | Au | B | Ba | Be | Bi | Cd | Co | Cr | Cu | Fe | Mn |
| 0 | < 0.01 | <20 | 1.7 | < 0,01 | 59 | 37 | < 0.1 | < 0.01 | < 0.01 | 0.03 | < 0.5 | 0.3 | <10 | <7 |
| 10 | < 0.01 | <20 | 1.4 | < 0,01 | 77 | 33 | < 0.1 | < 0.01 | < 0.01 | 0.03 | < 0.5 | 0.6 | <10 | <7 |
| 70 | < 0.01 | <20 | 1.9 | < 0,01 | 183 | 132 | < 0.1 | < 0.01 | < 0.01 | 0.06 | < 0.5 | < 0.2 | 25.1 | 424 |
| 130 | < 0.01 | <20 | 1.9 | < 0,01 | 388 | 379 | < 0.1 | < 0.01 | < 0.01 | 0.14 | < 0.5 | < 0.2 | 15.6 | 329 |
| 170 | 0.013 | <20 | 1.6 | < 0,01 | 430 | 411 | < 0.1 | < 0.01 | < 0.01 | 0.14 | < 0.5 | 0.6 | <10 | 304 |
| 230 | < 0.01 | <20 | 1.7 | < 0,01 | 566 | 529 | 0.2 | < 0.01 | < 0.01 | 0.18 | < 0.5 | < 0.2 | <10 | 263 |
| 290 | 0.013 | 30.6 | 2.7 | < 0,01 | 771 | 671 | 0.42 | < 0.01 | < 0.01 | 0.4 | 0.5 | < 0.2 | 74.2 | 268 |
| 350 | 0.016 | 34.8 | 3.2 | < 0,01 | 941 | 675 | 0.42 | < 0.01 | < 0.01 | 0.52 | 0.8 | 0.2 | 87.7 | 274 |
| 410 | 0.018 | 26.3 | 3 | < 0,01 | 595 | 686 | 0.6 | < 0.01 | < 0.01 | 0.53 | 0.7 | < 0.2 | 63.9 | 271 |
| 450 | < 0.01 | <20 | 2.3 | < 0,01 | 491 | 483 | 0.22 | < 0.01 | < 0.01 | 0.33 | < 0.5 | < 0.2 | 15.2 | 265 |





**Table 1 (continued for more elements)**

| depth | Mo | Ni | Pb | Rb | Sb | Se | Sn | Sr | Ti | Tl | U | V | Y | Zn |
|---|---|---|---|---|---|---|---|---|---|---|---|---|---|---|
| 0 | 1.4 | 0.3 | 0.2 | 50 | 0.04 | 0.1 | < 0.1 | 200 | < 1.0 | < 0.01 | 0.8 | 1.7 | < 0.5 | < 9 |
| 10 | 0.4 | 0.9 | 0.6 | 61 | 0.06 | 0.1 | 0.2 | 717 | 1.6 | < 0.01 | 0.7 | 2 | < 0.5 | < 9 |
| 70 | 0.4 | 0.9 | 0.2 | 60 | 0.01 | 0.1 | < 0.1 | 717 | 1.4 | < 0.01 | 0.68 | 2.1 | < 0.5 | < 9 |
| 130 | 0.7 | 0.8 | 0.1 | 109 | 0.3 | < 0.1 | < 0.1 | 2730 | 2.5 | < 0.01 | 0.32 | 3.1 | < 0.5 | < 9 |
| 170 | 0.9 | 1 | 0.3 | 104 | 0.02 | 0.1 | 0.4 | 2740 | 2.9 | < 0.01 | 0.25 | 2.9 | < 0.5 | < 9 |
| 230 | 1.3 | 1 | 0.1 | 128 | 0.01 | 0.1 | 0.2 | 3790 | 4.2 | < 0.01 | < 0.10 | 3.3 | < 0.5 | < 9 |
| 290 | 0.7 | 2 | 0.1 | 192 | 0.04 | 0.2 | < 0.1 | 5540 | 1.7 | < 0.01 | < 0.10 | 5.4 | < 0.5 | < 9 |
| 350 | 0.6 | 3.1 | 0.1 | 209 | 0.01 | 0.2 | < 0.1 | 5750 | 5.7 | < 0.01 | < 0.10 | 7.2 | < 0.5 | < 9 |
| 410 | 0.5 | 4 | 0.1 | 205 | 0.02 | 0.2 | 0.1 | 5650 | 8.8 | < 0.01 | < 0.10 | 7.1 | < 0.5 | < 9 |
| 450 | 0.8 | 1.8 | 0.1 | 164 | 0.01 | 0.2 | 0.3 | 4250 | 7.7 | < 0.01 | 0.26 | 5.1 | < 0.5 | 21 |
| Dis-solved | Mo | Ni | Pb | Rb | Sb | Se | Sn | Sr | Ti | Tl | U | V | Y | Zn |
| 0 | 1.5 | < 0.3 | < 0.04 | 48 | 0.05 | 0.2 | < 0.1 | 197 | < 1.0 | < 0.01 | 0.85 | 1.6 | < 0.5 | <9 |
| 10 | 1.3 | < 0.3 | 0.1 | 48 | 0.03 | 0.2 | < 0.1 | 191 | < 1.0 | < 0.01 | 0.81 | 1.8 | < 0.5 | <9 |
| 70 | 0.4 | < 0.3 | < 0.04 | 58 | 0.01 | < 0.1 | < 0.1 | 726 | < 1.0 | < 0.01 | 0.7 | 2 | < 0.5 | <9 |
| 130 | 0.3 | 0.5 | < 0.04 | 93 | 0.01 | 0.1 | < 0.1 | 2330 | 2.3 | < 0.01 | 0.33 | 3 | < 0.5 | <9 |
| 170 | 0.4 | 0.5 | < 0.04 | 97 | 0.01 | 0.1 | < 0.1 | 2570 | 2.1 | < 0.01 | 0.26 | 2.9 | < 0.5 | <9 |
| 230 | 0.9 | 0.7 | < 0.04 | 113 | < 0.01 | 0.1 | < 0.1 | 3360 | 3.2 | < 0.01 | 0.14 | 3.2 | < 0.5 | <9 |
| 290 | 0.3 | 1.6 | < 0.04 | 165 | 0.02 | 0.2 | < 0.1 | 4760 | 7.9 | < 0.01 | < 0.10 | 5.4 | < 0.5 | <9 |
| 350 | 0.3 | 2.4 | < 0.04 | 187 | < 0.01 | 0.2 | < 0.1 | 5120 | 11.5 | < 0.01 | < 0.10 | 6.8 | < 0.5 | <9 |
| 410 | 0.3 | 3.1 | < 0.04 | 198 | 0.02 | 0.2 | < 0.1 | 5430 | 12.5 | < 0.01 | < 0.10 | 7.2 | < 0.5 | <9 |
| 450 | 0.9 | 1.1 | < 0,04 | 140 | < 0.01 | 0,1 | < 0.1 | 3630 | 6.4 | < 0.01 | 0.25 | 5.1 | < 0.5 | <9 |



**Table 2: main ions and nutrient concentration in Lake Kivu; all concentrations are in [mg/l]; depth in [m of rope], which to our experience was 6% higher that real depth. Water was sampled from platform (GIS) on 8.03.2018; only the depths 410 m and 450 m were sampled from the boat (DEEP) on 12.03.2018.**

| Depth | K | Na | Mg | Ca | SO₄ | Cl | TP | TNb | TOC |
|---|---|---|---|---|---|---|---|---|---|
| 0 | 78.6 | 102 | 77.6 | 9.63 | 21.4 | 25.1 | 0.01 | < 0.08 | 1.71 |
| 10 | 79.3 | 102 | 78.1 | 9.54 | 22 | 25.7 | 0.006 | < 0.08 | 1.25 |
| 70 | 98.5 | 130 | 101 | 28.5 | 14.9 | 31.8 | 0.18 | 2.93 | 1.73 |
| 130 | 154 | 210 | 151 | 76 | 13.9 | 51.2 | 0.975 | 11 | 1.68 |
| 170 | 163 | 223 | 159 | 78.9 | 18.3 | 50.3 | 1.17 | 12.7 | 2.74 |
| 230 | 186 | 259 | 188 | 93.6 | 14 | 60.3 | 1,4 | 15.3 | 2.59 |
| 290 | 266 | 383 | 307 | 119 | 18 | 82.4 | 3.95 | 45.7 | 2.77 |
| 350 | 302 | 426 | 369 | 130 | 17.4 | 92.6 | 4.79 | 49.9 | 4.38 |
| 410 | 325 | 456 | 388 | 136 | 23.4 | 72 | 4.91 | 55.3 | 5.62 |
| 450 | 234 | 329 | 259 | 94.8 | 22.8 | 91.9 | 3 | 33.8 | 3.09 |

