# Peer review of "Reliable reference for the methane concentrations in Lake Kivu at the beginning of industrial exploitation"

_Hydrology and Earth System Sciences, 2019_

## Referee Comment (RC1) · Minoru Kusakabe (Referee) · 20 Jun 2019

This is a technically important paper for the study of Lake Kivu, since the lake contains a large amount of CH4 and CO2 dissolved in the deep water. The local community is anxious about a possible limnic eruption as happened in Lakes Nyos and Monoun in Cameroon in mid-80s. Regular and frequent monitoring of CH4 and CO2 concentrations in the lake is the only way to assess the possibility of limnic eruption in the future. For this reason, this manuscript supplies a reliable method for the gas measurement. Since the lake is located in a remote area of Africa, regular and frequent monitoring of CH4 and CO2 concentrations in the lake has to be carried out by local scientists.

[Figure]

I hope parts of the equipment are easily available and not too expensive for the local scientists.

The authors say that the possibility of limnic eruption at Lake Kivu is not high, because the CH4 profiles obtained by the authors do to show appreciable change when compared with those in the literatures. This view needs to be confirmed by further measurements, hopefully performed by the local scientists.

The manuscript contains analyses of trace elements in lake water. Unfortunately, such data do not play an important role in the current manuscript. The reviewer feels that the sction for trace elements is not necessary, and suggests to delete it from the manuscript. Probably this is the way to make the manuscript more impressive and effective for the authors' main objective.

The reviewer's comments were embedded in the manuscript using PDF's annotation function.

Please also note the supplement to this comment:
https://www.hydrol-earth-syst-sci-discuss.net/hess-2019-228/hess-2019-228-RC1-supplement.pdf

**Supplement:**

[revised manuscript text omitted]

---

## Author Comment (AC1) · 11 Jul 2019

Reply to https://doi.org/10.5194/hess-2019-228-RC1 Interactive comment on Lake Kivu paper by Minoru Kusakabe.

REPLY: We appreciate very much receiving a written comment by an expert on limnic gas loads as Prof. M. Kusakabe, who was a leading scientist during the investigations and the remediation activities at Lake Nyos and Lake Monoun.

COMMENT: This is a technically important paper for the study of Lake Kivu, since the lake contains a large amount of CH4 and CO2 dissolved in the deep water. The local

community is anxious about a possible limnic eruption as happened in Lakes Nyos and Monoun in Cameroon in mid-80s. Regular and frequent monitoring of $CH_4$ and $CO_2$ concentrations in the lake is the only way to assess the possibility of limnic eruption in the future. For this reason, this manuscript supplies a reliable method for the gas measurement.

REPLY: We thank for the kind words on the relevance of this investigation.

COMMENT: Since the lake is located in a remote area of Africa, regular and frequent monitoring of $CH_4$ and $CO_2$ concentrations in the lake has to be carried out by local scientists. I hope parts of the equipment are easily available and not too expensive for the local scientists.

REPLY: Other than in Lake Nyos, Lake Kivu contains a precious gas, and hence there is commercial interest also from public authorities to purchase equipment for the survey.

COMMENT: The authors say that the possibility of limnic eruption at Lake Kivu is not high, because the $CH_4$ profiles obtained by the authors do to show appreciable change when compared with those in the literatures. This view needs to be confirmed by further measurements, hopefully performed by the local scientists.

REPLY: Right, the measurements of 2018 do not indicate an imminent endangerment through limnic eruptions. However, the further observation is mandatory to guarantee safety also in future in particular during and because of the industrial exploitation. Besides the documentation of the current situation, the supply of measurement strategies for the survey was one of the central objectives of this investigation.

COMMENT: The manuscript contains analyses of trace elements in lake water. Unfortunately, such data do not play an important role in the current manuscript. The reviewer feels that the sction for trace elements is not necessary, and suggests to delete it from the manuscript. Probably this is the way to make the manuscript more impressive and effective for the authors' main objective.

REPLY: We agree fully; the gas story does not rely on the measurements of trace elements. Removing this section would shorten the paper. We also like compact and concise papers. However, as we want to document the situation at the start of industrial exploitation, we feel the chemical situation should remain part of the manuscript. Previous publications deviated from our measurements in several cases "noticeably" (as written in our manuscript). As we are convinced of the quality (reliability) of our measurements, we see the need to have them published in a respected journal. We would much rather refer to chemical measurements in HESS than in HESS-discussions. Other authors may not even appreciate our chemical measurements as much as previously published measurements. We hope the editor can follow our arguments and opts for retaining the chemical measurements.

COMMENT: The reviewer's comments were embedded in the manuscript using PDF's annotation function.

REPLY: We thank for the detailed comments embedded in the manuscript. We have checked them and we are confident to include most of them or find a compromise that is acceptable for the reviewer. We will do this, as soon as the second review is available towards the end of the discussion period.

for all authors Bertram Boehrer
* * *

---

## Short Comment (SC1) · 5 Aug 2019

Overall the authors have presented interesting data of Lake Kivu gas content that is needed for the evaluation of the threat those gases pose to the local communities and the potential energy resource.

Comment 1: It would be helpful to reader if the data displayed in figures 3 (Temperature, Conductivity, pH, DO), 4 ($CH_4/CO_2$ concentrations and partial pressures), and 6 ($CO_2$ concentrations) could also be displayed in tabular form in a supplementary document or in an online database.

[Figure]

Comment 2: The Figure 6 x-axis label should be CO2 not CH4

Comment 3: In section 4.3 a sentence states a quick calculation based on henry coefficients were made. In the methods section there is a reference to other manuscripts and pK values, but it would be helpful for the reader if the authors could further clarify either the Henry coefficients used in this work or improve the description of the methods used to calculate/estimate the partial pressure contributions to the total gas pressure.

Comment 4: The authors state that:

"The major contributors were methane, carbon dioxide and nitrogen. While we had good measurements of methane and carbon dioxide, we missed data of similar quality for nitrogen: hence, we could not give a proper calculation of gas pressure from gas concentrations."

Can the authors elaborate on what issues attributed to the missing data of nitrogen?

---

## Referee Comment (RC2) · Anonymous Referee #2 · 9 Aug 2019

The authors interested in the measurements of gases concentrations in very deep waters of Lake Kivu. This work is very interesting, since the latest studies were alarming and suggested that methane concentrations in Lake Kivu were increasing, and that a limnic eruption was possible in a near future. This study shows that gases concentrations, and especially methane concentrations, did not increase since the first measurements made by Schmitz and Tietze, and thus that methane production is not higher than methane oxidation+methane fluxes to the atmosphere. I think that this study is important and overall reliable, with good measurements methods. Actually, I decided to put this manuscript in "major revisions" mainly for grammatical and language reasons, and also because numerous specifications and details must be added.

In particular, I have some questions on M&M (please see below).

- P2 L3: "large amounts of dissolved methane" => please specify how much, the readers who do not work on Lake Kivu won't know - P2 L8-9: "The survey of the management prescription" => What does that mean ? - P2 L16: "an increasing risk of limninc eruption" - P2 L17-19: Sentence not clear, please reformulate - P2 L24: "All of them, even the more recent measurements trials, had to struggle with the loss of gas and water while recovering samples to the surface". - P2-3 L29-2: Sentence not clear, please reformulate. - P3 L14: separate what ? - P3 L23: dissolved solids => which ones? - P4 L5: "Map of Lake Kivu, showing the sampling sites (GIS and DEEP) and the area of the resource zone (below 260m). - P4 L10: "at a maximum depth of 410 m": not clear. Did you make only one depth (410 m) or the depth 410 m was the maximum one, and you also sampled other depths. Please be clearer about the depths sampled. You also sampled the deepest location DEEP, but there is not precision on the depth sampled at this location. So, which depths did you sample at the platform GIS and which depths did you sample at the location DEEP? - P5 L1: Also see the study of Borges et al (2011) for the horizontal variability/homogeneity - P6 L3: "Gas sampling device, with pump controller...." - P7 L8: "over night" => samples were not preserved with anything. Do you consider that any biological activity can occur over night in the water (methane production for example), and thus that the gas composition cannot change during the night? How can you stipulate this? Were the bags well kept in dark? Please give more details and specify. - P7 L9: What is the gas in the headspace? - P8 L8-11: Sentence not clear, please reformulate and clarify - P8 L20-24: This conclusion must not be in the M&M. Please move it in the discussion or conclusion sections. - P9 L4-6 : Sentence not clear, please reformulate - P9 L29-31: grammatically not correct, please reformulate. Also, what is a "compensation reservoir" ? - P10: The norm ISO is not a protocol. The reader needs more details of the analytical methods. Please well describe the steps of the analyses. - P10 L16: according to Koroleff ??? Reference? - P11 L4: According to DO vertical profiles, 2 field campaigns were conducted during the dry season (27/5/17 and 2/6/17) and 2 field campaigns during the rainy season (9/3/18 and 13/3/18). Why

did you choose to make measurements during both seasons? Indeed, this study focuses on very deep waters, where there is no influence of seasonality. Please clarify - P11 L7-10 : In the results section, you must not reference other studies. Referencing the literature is a discussion, not results - P11 L13: The resolution of the figure 3 is not good, especially the writing. Please improve. Also, please remove the minus in front of the depths - P11 L14: "against depth" => according to depth - P12 L1-2: Please add this limit on the graph - P12 L8: "Successfully" => What does that mean? - P 12 L9: The previous results used in this study should be specificied above (in the section 2.1). At this stage, it was not clear that the authors did not sample in 27/5 and 2/6 for this study. - P12 L10-11: Please better describe your results. Example: "CH4 concentrations reached 18 mmol/l at 450m", etc. - P13 L1: Please put letters for identify panels, it is easier for the reader (upper panel = A; middle panel = B, etc) - P13 L2: "from the platform GIS" - P13 L4: Until now, the authors did not use the terms "site 2" and "site 1", but "GIS" and "DEEP". Please stay constant - P14 L2: "greater" => "deeper" - P14 L5: What is the potential ressource zone? Please well define the different zones of the lake, and overall showing them on vertical profiles graphs - P14 L6-7: It is not a result, it is a discussion - P14 L8-9: It is not a result, it is a discussion - P15 L6: I don't understand "some variability of temperatures"; please clarify - P15 L8-9: I don't see how you can conclude this, only on the base on the "look" of the vertical profiles. Please detail - P15 L10-16: Nothing new, already well developped in all the papers on Lake Kivu. Please summarize - P15 L20-23: The studies are not properly referenced. Please reformulate; example: "... with previous observations (Schmitz and Kufferath 1952, Tietze 1974, Schmid et al 2005)" - P16 L5: Methane concentrations (mmol/l) - P17 L10: Carbon dioxide concentrations (mmol/l) - P18 L4: The study of Roland et al 2018 is the only study that quantified in situ methane oxidation in Lake Kivu. As the authors talk about methane oxidation, it would be correct to also reference this study - P18 L16-19: Already said above - P19 L15-26: I don't see what these data bring to the manuscript. They are unuseful for the purpose of the present study, and they are not well developed - P19 L16: "previous ones" - P19 L20: You did not study Kabuno Bay in

the present study - P19 L21-22: I agree with you, numerous errors are published in the study of Tassi et al 2009 - P19 L22-23: I don't understand why the authors talk about the limit for drinking water in Germany. It does not make sense in this study! - P20 L16 and L19: "on the base on" - P24: The design of the table is not clear and not beautiful. Please improve

---

## Author Comment (AC2) · 19 Aug 2019

Reply to short comment by Casey Quinn casey.quinn@colostate.edu

——

Overall the authors have presented interesting data of Lake Kivu gas content that is needed for the evaluation of the threat those gases pose to the local communities and the potential energy resource.

We appreciate very much that the relevance for science and local communities is em-

phasized.

————

Comment 1: It would be helpful to reader if the data displayed in figures 3 (Temperature, Conductivity, pH, DO), 4 (CH4/CO2 concentrations and partial pressures), and 6 (CO2 concentrations) could also be displayed in tabular form in a supplementary document or in an online database.

We are very happy to share our measurements once the manuscript has been accepted for publication and proper referencing is possible. We also appreciate the request, as it shows that the presented measurements are relevant and will be used and cited. It also confirms that the data and the measurement techniques will form the reference we intend to produce. In the final version in HESS, we can include the tabled values in a supplement, if this is requested. The display of values of probe profiles is not possible as each profile has around 10000 lines.

————

Comment 2: The Figure 6 x-axis label should be CO2 not CH4

This is correct, thanks for pointing this out. We will do the correction.

————

Comment 3: In section 4.3 a sentence states a quick calculation based on henry coefficients were made. In the methods section there is a reference to other manuscripts and pK values, but it would be helpful for the reader if the authors could further clarify either the Henry coefficients used in this work or improve the description of the methods used to calculate/estimate the partial pressure contributions to the total gas pressure.

The conversion of gas concentration into gas pressures is complex. However, we felt a connection between the measured gases and gas pressures should be presented to prove that the data are consistent. On purpose we did not include details as we were

aware that the conversion could be done more accurately, but this was too complex to be included in this manuscript. We will NOT present coefficients which are not optimal for calculating gas pressures. Henry coefficients for methane, carbon dioxide and nitrogen at atmospheric pressures are easily available in the literature (e.g. Sander et al 1999) also including temperature dependence. We rather refer to Bärenbold et al, a manuscript that has been submitted a few days ago, which shows what is believed to be the most accurate way to convert concentrations to gas pressures for Kivu conditions.

———-

Comment 4: The authors state that: "The major contributors were methane, carbon dioxide and nitrogen. While we had good measurements of methane and carbon dioxide, we missed data of similar quality for nitrogen: hence, we could not give a proper calculation of gas pressure from gas concentrations." Can the authors elaborate on what issues attributed to the missing data of nitrogen?

The sampling technique aimes at getting CH4 and CO2 concentrations measured reliably and accurately. No doubt, N2 measurements are important, but they were not the topic of this investigation. The difficulty with measuring N2 is the relatively low concentration: The forming headspace in the sampling bags consists mainly of CO2 and CH4. N2 (and H2O) form a small portion. Hence also the N2-peak in each GC diagramme is small and hence the relative error is high. In addition, air consists mainly of N2 (and O2 and little Ar). Hence a small contamination affects the N2 measurement enormously, while the effect on CH4 or CO2 is marginal. In general, the sampling bag approach is also suited for N2 quantification as shown in Horn et al, 2016. However it requires further effort to quantify the contribution of air contamination and measurement accuracy of the GC before such samples can be used for a reliable N2 quantification.

---

## Author Comment (AC3) · 22 Aug 2019

Completing the final response , we also upload the pdf file commented by Kusakabe, where we indicate how we intend to include the hints in a new version of the manuscript for HESS.

There is no issue remaining, except that we are still convinced that the chemical data should be included in the proper paper. Admittedly, the methane is the attractive story and this update (plus instrumental recommendation) should warrant a publication , but for documenting the current situation the chemical situation should be included, especially as the most recent publication on the chemistry of Lake Kivu needs to be

urgently replaced by better data.

Our answers on the major points are still valid as written in the first reply.

We would look forward to see our work properly published in HESS.

for all authors Bertram Boehrer

Please also note the supplement to this comment:
https://www.hydrol-earth-syst-sci-discuss.net/hess-2019-228/hess-2019-228-AC3-supplement.pdf

---

## Author Comment (AC4) · 22 Aug 2019

Reply to anonymous Referee #2

——

The authors interested in the measurements of gases concentrations in very deep waters of Lake Kivu. This work is very interesting, since the latest studies were alarming and suggested that methane concentrations in Lake Kivu were increasing, and that a limnic eruption was possible in a near future. This study shows that gases concentrations, and especially methane concentrations, did not increase since the first measure-

ments made by Schmitz and Tietze, and thus that methane production is not higher than methane oxidation+methane fluxes to the atmosphere. I think that this study is important and overall reliable, with good measurements methods. »»»»» We appreciate the kind words: We also appreciate that the important consequences for political decisions have been recognized by the reviewer.

———

Actually, I decided to put this manuscript in "major revisions" mainly for grammatical and language reasons, and also because numerous specifications and details must be added.

» We definitely appreciate any comment that is suited to improve the quality and legibility of our paper. Below, we indicate briefly, how we intend to include all comments by anonymous referee #2 in a new final version.

» Only two of his/her points refer to the contents of the paper:

» (1) supposed methane production in a sampling bag over night: The methane production is not relevant. Estimates show that methane production over night may be in the range of 10E-7 of the amount of methane recovered from the bags.

» (2) inclusion of chemical data: The chemical data are part of of the documentation of the current status. The most recent publication on chemical composition (Tasssi et al) shows problems and must urgently be replaced. The chemical data should be included in the proper publication in HESS.

———-

Details:

In particular, I have some questions on M&M (please see below).  - P2 L3: "large amounts of dissolved methane" => please specify how much, the readers who do not work on Lake Kivu won't know »»»»» we will add the number of 40 km$^3$ below 260 m

- P2 L8-9: "The survey of the management prescription" => What does that mean ? »»»»» there have been management prescriptions issued, which the gas companies have to follow. They are subject to survey from time to time.

- P2 L16: "an increasing risk of limninc eruption" »»»»» Something went wrong in this citation. No such expression in the manuscript

- P2 L17-19: Sentence not clear, please reformulate »»»»» Not clear which sentence

- P2 L24: "All of them, even the more recent measurements trials, had to struggle with the loss of gas and water while recovering samples to the surface". »»»»» O.K. the reviewer has proposed to change the sentence into the version that he lists above: it will be accepted.

- P2-3 L29-2: Sentence not clear, please reformulate. »»»»» We will reformulate this sentence

- P3 L14: separate what ? »»»»» "in separate" means "separately"

- P3 L23: dissolved solids => which ones? »»»»» This is the introduction. This is not the place to include a list of measured dissolved solids.

- P4 L5: "Map of Lake Kivu, showing the sampling sites (GIS and DEEP) and the area of the resource zone (below 260m). »»»»» Proposed change of figure caption will be accepted and implemented.

- P4 L10: "at a maximum depth of 410 m": not clear. Did you make only one depth (410 m) or the depth 410 m was the maximum one, and you also sampled other depths. Please be clearer about the depths sampled. You also sampled the deepest location DEEP, but there is not precision on the depth sampled at this location. So, which depths did you sample at the platform GIS and which depths did you sample at the location DEEP? »»»»» No this is only the description of the sampling site (5km from Gisenyi at maximum depth 410m); no sampling depths are listed here.

- P5 L1: Also see the study of Borges et al (2011) for the horizontal variability/homogeneity »»»»»» Listing Borges et al here would be misleading, as they have done measurements in the surface layer only, and we refer to horizontal variability below 150m, but Borges et al is already in the references.

- P6 L3: "Gas sampling device, with pump controller...." »»»»»» o.k. change will be accepted and implemented in new version

- P7 L8: "over night" => samples were not preserved with anything. »»»»»» Right: no conservation as there is no contact with any new substance

(1) Do you consider that any biological activity can occur over night in the water (methane production for example), and thus that the gas composition cannot change during the night? How can you stipulate this? Were the bags well kept in dark? Please give more details and specify.

» It is dark inside the bags, as they have an aluminium layer coated with plastic. In addition, there was no light in the laboratory during the night. While sampling, the bags were kept out of direct sun light. In previous experiments, we had tested the behaviour of samples similar to Kivu over one fortnight and we did not see any changes, neither by production nor by diffusive losses out of the bags (see Horn et al. 2016).

» Of course methane production and methane removal also happens in pelagic waters, although most of the chemical transformation happens at sediment surfaces. We used new bags, hence there was no a-priori infection with methane producing bacteria; in conclusion we expect small rates.

» In Lake Kivu deep water, methane production and removal (oxidation) together with diffusive processes nearly balance each other. Time scales of change are at least 100 to 1000 years. Cutting the diffusive connection (by sampling), would make us expect a relative change in a similar order of magnitude, i.e. $1/(2*365*300) \sim 5*10E-6$ over night in a first order approach estimate.

» In addition we offer a brief calculation: 1 Liter of Kivu water produces 1Liter of gas, i.e. roughly 250 ml ∼ 0.01 mol of CH4; Rates of methane production from pelagic waters were measured by Iversen et al. (1987; Limnol Oceanogr. 32(4), 804-814 in Big Soda Lake). Rates lie in the range of few nmol/(l*day) and can produce a portion of 10E-7 of the amount of CH4 already in the bag over night.. Of course, conditions in Lake Kivu can be remarkably different to Great Soda Lake but very unprobably this much. The reviewer should have listed where he took the huge numbers for his assumptions from.

» Using these estimates and the test of methane samples in the sampling bags indicate that there are no noticeable effects from storing samples over night.

- P7 L9:  What is the gas in the headspace?   »»»»» We are not using the word "headspace" here. The "headspace" results from degassing. No additional gases have been added to the sampling bags.  An addition of headspace is only required, if the sample does not produce enough gas to warrant a reliable measurement; this is not needed (and would not be helpful) in Lake Kivu deep water.

- P8 L8-11: Sentence not clear, please reformulate and clarify »»»»» We will edit this sentence, as requested by the reviewer

- P8 L20-24: This conclusion must not be in the M&M. Please move it in the discussion or conclusion sections.  »»»»» The wording "in conclusion" does not indicate that we have a major conclusion of the manuscript placed in the wrong section, but we will check this paragraph and modify it accordingly.

- P9 L4-6 :  Sentence not clear, please reformulate »»»»» The sentence reads: "If the total gas pressure inside the gas space equals the outside pressure (hydrostatic plus atmospheric pressure), a virtual gas bubble withstands the pressure at depth and persists long enough to start moving upwards through the water column through its own buoyancy." – It is not clear, what the difficulty is for the reviewer.

- P9 L29-31: grammatically not correct, please reformulate. Also, what is a "compensation reservoir" ? »»»»» o.k. we will deal with this sentence again.

- P10: The norm ISO is not a protocol. The reader needs more details of the analytical methods. Please well describe the steps of the analyses. »»»»» The word protocol is not used on page 10. On purpose, our description of analytical methods is very brief, because they do not form the core of this paper. However the included material must be sufficient to explain procedures and methods. DIN and ISO are clearly referred to: also the reviewer does not explicitly mention a missing detail. Hence we will check the section for completeness carefully again.

- P10 L16: according to Koroleff ??? Reference? »»»»» We will complement this reference in the final version of the paper

- P11 L4: According to DO vertical profiles, 2 field campaigns were conducted during the dry season (27/5/17 and 2/6/17) and 2 field campaigns during the rainy season (9/3/18 and 13/3/18). Why did you choose to make measurements during both seasons? Indeed, this study focuses on very deep waters, where there is no influence of seasonality. Please clarify »»»»» The reviewer is entirely right! There is no seasonality in the deep waters. Other constraints were more important for the choice of sampling dates.

- P11 L7-10 : In the results section, you must not reference other studies. Referencing the literature is a discussion, not results »»»»» o.k if this is bothering the reviewer, we will find another place to list references on meromixis.

- P11 L13: The resolution of the figure 3 is not good, especially the writing. Please improve. Also, please remove the minus in front of the depths - P11 L14: "against depth" => according to depth »»»»» Right! there will be a better resolution figure for the final version

- P12 L1-2: Please add this limit on the graph »»»»» We will find a place to display "ressource zone" and "potential ressource zone" in a display.

- P12 L8: "Successfully" => What does that mean? »»»»» We agree that this is an unfortunate expression. We will remove the word "successfully", as samples from unsuccessful sampling do not exist.

- P 12 L9: The previous results used in this study should be specificied above (in the section 2.1). At this stage, it was not clear that the authors did not sample in 27/5 and 2/6 for this study. »»»»» We cannot really follow where the reviewer has difficulties. We clearly list the sampling dates here. The reference to section 2.1 Lake Kivu map / location does not make sense.

- P12 L10-11: Please better describe your results. Example: "CH4 concentrations reached 18 mmol/l at 450m", etc. »»»»» If the reviewer wishes a few descriptive sentences about the findings here, we can add them easily ( we leave this to the editor)

- P13 L1: Please put letters for identify panels, it is easier for the reader (upper panel = A; middle panel = B, etc) »»»»» In more complex depictions, we label the panels, but here, it is easier to refer to upper , middle and lower panel. We believe readers can easily distinguish (editor's choice).

- P13 L2: "from the platform GIS" - P13 L4: Until now, the authors did not use the terms "site 2" and "site 1", but "GIS" and "DEEP". Please stay constant »»»»» We will correct this to remain consistent.

- P14 L2: "greater" => "deeper" »»»»» Writing "at depth deeper than .." is colloquial English. It raises the question whether depth itself can be deep. We propose to change the sentence to "at depths below 400m."

- P14 L5: What is the potential ressource zone? Please well define the different zones of the lake, and overall showing them on vertical profiles graphs »»»»» As mentioned above we will include the depth of ressource zone and potential ressource zone in a display.

- P14 L6-7: It is not a result, it is a discussion »»»»» We will change the sentence
- P14 L8-9: It is not a result, it is a discussion »»»»» We will change the sentence

- P15 L6: I don't understand "some variability of temperatures"; please clarify »»»»» We will change the sentence

- P15 L8-9: I don't see how you can conclude this, only on the base on the "look" of the vertical profiles. Please detail »»»»» We do not write "look" and we do not conclude anything in these two lines.

- P15 L10-16: Nothing new, already well developped in all the papers on Lake Kivu. Please summarize »»»»» Correct we agree fully! We confirmed these facts for the years 2017 and 2018, as we wrote.

- P15 L20-23: The studies are not properly referenced. Please reformulate; example: "... with previous observations (Schmitz and Kufferath 1952, Tietze 1974, Schmid et al 2005)" »»»»» These are not citations, these are listings of investigators and years of sample collection.

- P16 L5: Methane concentrations (mmol/l) »»»»» we will replace symbols at figure axes by words where advisable.

- P17 L10: Carbon dioxide concentrations (mmol/l) »»»»» Right! we will replace the label at the x-axis.

- P18 L4: The study of Roland et al 2018 is the only study that quantified in situ methane oxidation in Lake Kivu. As the authors talk about methane oxidation, it would be correct to also reference this study »»»»» We were not aware of the Roland paper; we will cite it.

- P18 L16-19: Already said above »»»»» We are not repeating the statement, but refer to the response time to discuss the posssibilities that derive from this faster instrument. We think about what needs to be changed here.

(2) - P19 L15-26: I don't see what these data bring to the manuscript. They are unuseful for the purpose of the present study, and they are not well developed »»»»» We clearly oppose the reviewer's statement. He may say that the data do not fit into the focus oft he manuscript, but calling them "unuseful" is not correct. These data are very useful for anybody who needs reliable chemical data from Lake Kivu. These data show clear deviations from Tassi et al. and hence form a very important reference. We do not see any other chemical analysis of Kivu pelagic water of a similar quality. The chemical data also form part of the definition of the situation in the lake at the beginning of the industrial exploitation. We expect that in a few years time documented values of chemical composition in the literature will be highly valued. The chemical values are part of the reference and it is correct to list them together with the gas measurements. We do not see why this publication should exclude any important results, and why HESS should not receive the citations.

- P19 L16: "previous ones" »»»»» We will add "previous publications" in stead of "previous ones"

- P19 L20: You did not study Kabuno Bay in the present study »»»»» Right. We will remove the citation of Kabuno Bay

- P19 L21-22: I agree with you, numerous errors are published in the study of Tassi et al 2009 »»»»» (see also above) If the reviewer sees numerous errors in Tassi et al., he / she should encourage and support a new publication of chemical data in a respected journal like HESS to offer a better reference to the science community but also to the local decision takers.

- P19 L22-23: I don't understand why the authors talk about the limit for drinking water in Germany. It does not make sense in this study! »»»»» We are not talking about the limit for drinking water in Germany! We are talking about concentration of chromium in Lake Kivu, and compare this to an internationally respected reference for drinking water. The German limits are familiar to us. We would not know of another reference that would be suited noticeable better.

- P20 L16 and L19: "on the base on" »»»»» No! Proposed expression "on the base on"
is wrong. we retain the correct expression "on the base of"

- P24: The design of the table is not clear and not beautiful. Please improve »»»»» We
will make the table "more beautiful"!

---

## Editor Decision (ED1)

Dear Dr. Boehrer

Thank you for providing detail responses to the two reviews and the comments by C. Quinn during the interactive discussion.

I mostly agree with your replies and explanations and suggest that you revise the manuscript accordingly.

However, there is the issue of the trace metals where I do not follow your arguments but support the views of both reviewers.

First, both reviewers criticised that the trace metal topic wasn't related to the methane story and you share this view (Your response to Rev. 1: "We agree fully; the gas story does not rely on the measurements of trace elements. Removing this section would shorten the paper. ")

To include the trace metal results despite this disconnect to the core topic of the manuscript you argue as follows: "However, as we want to document the situation at the start of industrial exploitation, we feel the chemical situation should remain part of the manuscript. Previous publications deviated from our measurements in several cases "noticeably" (as written in our manuscript). As we are convinced of the quality (reliability) of our measurements, we see the need to have them published in a respected journal."

This is not convincing for including the results into a paper devoted to another topic. If the trace metal issue seems of sufficient importance you should consider to present it in a dedicated manuscript. This is essential for adequate scientific quality and peer review. Such a procedure ensures a proper context including presenting the state-of-the-art regarding these trace metals and possible deficiencies in available data. It also will influences the selection of reviewers.

For these reasons, I fully support the comments by the two reviewers and ask for removing the trace metal part from the manuscript.

Below I additionally list a few specific comments where I have specific recommendations (including those aspects where you explicitly asked for my opinion).

Comments related to the comment by C. Quinn (hess-2019-228-AC2)

Comment 1: Data sharing

Editor comment: Making data available is of essential importance. This is also clearly stated in data policy of HESS, see:

https://www.hydrology-and-earth-system-sciences.net/about/data_policy.html

You may make the data available through Supplementary information (e.g., in tabular form) but also use recognised data repositories (see e.g., https://repositoryfinder.datacite.org/) . Given these repositories I cannot follow your argument "The display of values of probe profiles is not possible as each profile has around 10000 lines.". Repositories can handle such data sets.

Please make the data sets available as requested by C. Quinn including the probe profiles unless you can provide convincing arguments that this is not feasible.

Comment 3: Details on pressure estimates

Author response : « Response "However, we felt a connection between the measured gases and gas pressures should be presented to prove that the data are consistent." "We will NOT present coefficients which are not optimal for calculating gas pressures."

Editor comment: If you provide numerical values in the manuscript for testing the plausibility of your results (comparing measured and estimated pressures) it is not acceptable to decline the wish of a reader to present the assumptions (incl. numerical values) on which your result are based.

Whether or not such a description has to go into the manuscript is a different question. However, I expect that you provide more information about your calculation (incl. the Henry coefficient used) in your response.

Comments related to the specific responses to M. Kusakabe (hess-2019-228-AC3-supplement)

p.3, L. 13 – 14: Please refer explicitly to Bärenbold et al.

p. 12, L. 2: Response: "It is not the purpose of this paper"

Editor comment: This holds true. Nevertheless, this piece of information may be of interest to readers and should be given if it's possible to do so in a one sentence.

p. 20, L. 21: Response: "beyond the scope of this paper."

Editor comment: This may be true, but it may help readers who are not so familiar with the topic to know about the basic process. Therefore, I suggest to simply add a sentence giving a short explanation (biogenic methane production in the sediment) and refer to respective literature.

Comments related to Rev. 2 (hess-2019-228-AC4)

Reviewer: "P2 L17-19: Sentence not clear, please reformulate"

Author response: "Not clear which sentence"

Editor: There is only one sentence here: *To avoid any endangerment for the population, a management prescription for withdrawal depth and deposition of partially degassed deep water and*

*wash water needed to be developed to avoid damage to the lake ecology and endangerment of the local population.* However, I consider the sentence as sufficiently clear.

Reviewer: "P11 L7-10 : In the results section, you must not reference other studies. Referencing the literature is a discussion, not results"

Author response: "o.k if this is bothering the reviewer, we will find another place to list references on meromixis."

Editor: If you can find another meaningful place to mention that the lake is meromictic, it's fine. However, you can also leave it as it is because the statement and the references are not part of a discussion but provide an explanation for proper understanding the results.

Reviewer: "P12 L10-11: Please better describe your results. Example: "CH4 concentrations reached 18 mmol/l at 450m"

Author response: "If the reviewer wishes a few descriptive sentences about the findings here, we can add them easily ( we leave this to the editor)"

Editor:  It is indeed a matter of taste. I suggest adding a few sentences to put the values into a broader context such that readers who are not very familiar with the topic can better appreciate the findings

Reviewer: "P13 L1: Please put letters for identify panels, it is easier for the reader (upper panel =A; middle panel = B, etc)

Author response: "In more complex depictions, we label the panels, but here, it is easier to refer to upper , middle and lower panel. We believe readers can easily distinguish (editor's choice)."

Editor: Again, this very much a matter of taste. It can remain as it is.